# Exogenous Melatonin Activating Nuclear Factor E2-Related Factor 2 (Nrf2) Pathway via Melatonin Receptor to Reduce Oxidative Stress and Apoptosis in Antler Mesenchymal Stem Cells

**DOI:** 10.3390/molecules27082515

**Published:** 2022-04-13

**Authors:** Huansong Jing, Xuyang Sun, Mengqi Li, Jingna Peng, Xiaoying Gu, Jiajun Xiong

**Affiliations:** Key Lab of Agricultural Animal Genetics, Breeding and Reproduction of Ministry of Education, College of Animal Science and Technology, Huazhong Agricultural University, Wuhan 430070, China; 15071472132@189.cn (H.J.); sunxuyangabc@163.com (X.S.); lx372328@163.com (M.L.); peng980906@163.com (J.P.); gxy19970310@163.com (X.G.)

**Keywords:** melatonin, antler, MSCs, oxidative stress, Nrf2 pathway, apoptosis

## Abstract

Antler growth depends on the proliferation and differentiation of mesenchymal stem cells (MSCs), and this process may be adversely affected by oxidative stress. Melatonin (MLT) has antioxidant functions, but its role in Cervidae remains largely unknown. In this article, flow cytometry, reactive oxygen species (ROS) identification, qPCR, and other methods were used to investigate the protective mechanism of MLT in H_2_O_2_-induced oxidative stress of antler MSCs. The results showed that MLT significantly increases cell viability by relieving the oxidative stress of antler MSCs. MLT inhibits cell apoptosis by protecting mitochondrial function. We blocked the melatonin receptor with luzindole (Luz) and found that the receptor blockade significantly increases H_2_O_2_-induced hyperoxide levels and causes significant inhibition of mitochondrial function. MLT treatment activates the nuclear factor E2-related factor 2 (Nrf2) antioxidant signaling pathway, up-regulates the expression of NAD(P)H quinone oxidoreductase 1 (*NQO1*) and other genes and it could inhibit apoptosis. In contrast, the melatonin receptor blockade down-regulates the expression of Nrf2 pathway-related genes, but significantly up-regulates the expression of apoptotic genes. It was indicated that MLT activates the Nrf2 pathway through the melatonin receptor and alleviates H_2_O_2_-induced oxidative stress and apoptosis in antler MSCs. This study provides a theoretical basis for further studying the oxidative stress and antioxidant process of antler MSCs and, thereby, increasing antler yields.

## 1. Introduction

Under natural conditions, the antler is shed and regenerated yearly [1]. The antler is the only mammal organ that can periodically regenerate [2]; therefore, it is a good model to study the regeneration of amputated limbs in mammals [3]. Antler growth depends on the proliferation and differentiation of mesenchymal stem cells (MSCs) in the antlerogenic periosteum (AP) [4,5]. In the process of cultivation, stress factors, including fine particulate matter (PM2.5) [6], ultraviolet light [7], and zearalenone in silage [8], may induce oxidative stress in antler tissues. Oxidative stress can cause cell damage and inhibit cell proliferation [9,10]. Additionally, ROS accumulation damages mitochondria, resulting in a decrease of mitochondrial membrane potential (MMP) and 5′-Adenylate triphosphate (ATP) levels, thereby, releasing cytochrome C (Cyt C) into the cytoplasm to activate the caspase (CASP) enzyme cascade, eventually inducing cell apoptosis [11,12]. Since oxidative stress ultimately affects antler yield, it is of great significance to study the oxidative stress response and the protective effects of melatonin on MSCs to increase yields.

Melatonin (MLT) is mainly secreted by the pineal gland [13] and plays an important role in the biological clock [14] and anti-tumor and anti-aging [15] processes. MLT can bind G-protein-coupled receptors and activate a variety of signaling pathways [16]. With its strong antioxidant properties, MLT can remove excessive free radicals and resist oxidation-induced damage in many tissues [17]. MLT can maintain optimal MMP and promote mitochondrial biogenesis [18]. MLT has been reported to protect cochlear hair cells from apoptosis and oxidative stress induced by nicotine [19] and it can protect ovarian cells from apoptosis and oxidative stress induced by benzopyrene [20]. Whether it can relieve oxidative stress in antler MSCs and its corresponding mechanisms need to be further investigated.

The activation of the nuclear factor E2-related factor 2 (Nrf2) pathway has been reported to inhibit oxidative stress [21]. Under normal circumstances, Kelch-like ECH-associated protein 1 (Keap1) and Nrf2 are combined in the cytoplasm, thus, remaining in the inactive state. When stimulated, Nrf2 is released from the Keap1–Nrf2 complex, transported to the nucleus, and bound with ARE (Au-rich element) to activate the expression of downstream target genes, such as oxygenase-1 (*HO-1*) and NAD(P)H quinone oxidoreductase 1 (*NQO1*), thus, inhibiting oxidative stress [22]. Recently, the effects of melatonin on the Nrf2 pathway have attracted increasing attention. Studies showed that melatonin can activate the Nrf2 pathway and reduce ROS accumulation to inhibit oxidative stress [23]. Furthermore, melatonin can protect lung cells from oxidative stress damage by activating the Nrf2 pathway [24]. 

Based on the literature reviewed above, this article evaluated the inhibitory effect of MLT on oxidative stress of antler MSCs and its protective effect on mitochondrial function. Luzindole (Luz) was used to block melatonin receptors (MTR) and then the phenotype of oxidative stress and the expression of Nrf2-related genes were investigated. We aimed to reveal the antioxidant effect and mechanism of MLT in the MSCs of antlers. Our results will contribute to the improvement of antler yield. 

## 2. Materials and Methods

### 2.1. Samples

Antlers were provided by the Jinsanxin Deer Farm, Caidian District, Wuhan city. The deer that provided the antler samples were *Cervus nippon* (Shuangyang sika deer)*,* aged 10–15 years. On the 45–60th day of antler growth, the cusp of the antler was cut off, and the mesenchymal cells of the antler were isolated by previously reported methods [25]. Cell type identification refers to expression level of pluripotency marker and MSC surface markers [26].

### 2.2. Culture of Cells

Cells were cultured at 37 °C, 5% CO_2_, and 95% air in DMEM/high glucose (HyClon, GE Healthcare, Logan, UT, USA, SH30022.01) containing 1% mycillin (Millipore, Burlington, MA, USA, MTS-AB2) and 10% fetal bovine serum (GIBCO, Waltham, MA, USA, 10270106).

### 2.3. Cell Treatments

The 96-well plates were inoculated with 5000 cells per well, and 6-well plates were inoculated with 2 × 10^5^ cells per well. MSCs were treated with H_2_O_2_ (0–1000 μmol/L with a gradient of 100 μmol/L) for 2 h to construct the oxidative stress model. For the MLT protection effect determination, MSCs were pretreated with MLT at the concentrations of 0, 1, 10, 100, and 1000 ng/mL (Meilunbio, Dalian, China, MB1475-S) for 12 h, followed by treatment with H_2_O_2_ (400 μmol/L) for 2 h. To reveal MLT protection mechanisms, MSCs were treated with 10 μmol/L luzindole (Sigma, Burlington, MA, USA, L2407) for 2 h, followed by 12 h MLT treatment (100 ng/mL) and 2 h H_2_O_2_ treatment (400 μmol/L).

### 2.4. Detection of Cell Viability 

The cell viability was determined with a cell proliferation toxicity assay kit (CCK-8 method, DOJINDO, Kumamoto, Japan, D3100L4054). After cell treatment, the culture medium was removed, and 100 μL CCK-8 working solution was added to each well (96-well plate). MSCs were cultured for 4 h at 37 °C, and the absorbance was determined at 450 nm.

### 2.5. Detection of Intracellular ROS Levels

The level of intracellular ROS was detected with an ROS detection kit (Biosharp, Beijing, China, BL714A). After cell treatment, the probe was loaded, and MSCs were incubated at 37 °C for 20 min. After washing with DMEM three times, fluorescence intensity was determined at 525 nm.

### 2.6. Detection of Intracellular MDA Levels

A malondialdehyde (MDA) content detection kit (Solarbio, Beijing, China, BC0025) was used to detect malondialdehyde levels. The cells from six-well plate cultures were digested with trypsin, ultrasonically broken, then added to the working solution and centrifuged to obtain the supernatant. The mixture was prepared and kept at 100 °C for 60 min. The absorbance of samples at 450 nm, 532 nm, and 600 nm was determined.

### 2.7. Detection of MMP Levels

Intracellular mitochondrial membrane potential was detected with an MMP kit (JC-1, Solarbio, Beijing, China, M8650). After cell treatment, the probe was loaded, and MSCs were incubated at 37 °C for 20 min. The cells were washed with JC-1 staining buffer (1×) twice. The fluorescence intensity was detected at 590 nm and 525 nm.

### 2.8. Detection of ATP Levels

A Celltiter-glo^®^ Luminescent Cell Viability Assay kit (Promega, Beijing, China, 0000447560) was used to determine the intracellular ATP levels. After cell treatment, the culture medium was removed from the wells, and the cells were washed once with DMEM. Subsequently, 100 μL of medium and 100 μL of reagent were added to each well and mixed in a shaking table for 2 min. The cells were incubated at room temperature for 10 min. Finally, the luminescence intensity was detected with a multi-function microplate tester.

### 2.9. Cell Cycle Analysis with Flow Cytometry

The cell cycle was detected with a DNA content assay kit (Solarbio, Beijing, China, CA1510). After cell treatment, the cells were collected, fixed with 70% ethanol for 12 h, resuspended with 100 μL RNase A, and bathed in 37 °C water for 30 min. The cells were stained with PI and incubated at 4 °C for 30 min. The cell cycle was analyzed with flow cytometry.

### 2.10. Cell Apoptosis Analysis with Flow Cytometry

An Annexin V-FITC/PI double staining apoptosis assay kit (Biosharp, Beijing, China, BL107A) was used to detect cell apoptosis. MSCs were treated with 3 replicates per treatment under different conditions. After treatment, the cells were collected in a centrifuge tube and stained with FITC dye and/or PI dye. Apoptosis was detected with flow cytometry with 10,000 cells per sample.

### 2.11. qPCR

Total RNA was extracted using the Trizol method, and cDNA was obtained by reverse transcription using HiScript II Q RT SuperMix qPCR kit (+gDNA Wiper, Vazymc, Nanjing, China, R223-01). DNA was extracted with an Animal Tissue/Cell Genomic DNA Extraction Kit (Solarbio, Beijing, China, D1700). The 2×Sybr Green qPCR Mix reagent (Aidlab biotechnologies CO. Ltd., Beijing, China, PC3301) was used for qPCR to detect the expression of related genes. The mixture system (10.0 μL) contained 5.0 μL mix, 1.0 μL primer (Table 1), and 4.0 μL diluted cDNA or DNA. The relative expression of the glyceraldehyde-3-phosphate dehydrogenase (GAPDH) gene was analyzed by the 2^−ΔΔCt^ method.

### 2.12. Western Blot

Total proteins were isolated using cell RIPA lysis buffer (Solarbio, Beijing, China, R0020). The extracted protein concentration was determined with an enhanced BCA protein assay kit (Solarbio, Beijing, China, PC0020). Afterward, samples containing 20 μg of protein were separated by sodium dodecyl sulfate polyacrylamide gel electrophoresis (Shanghai Yase Biomedical Technology Co., LTD, Shanghai, China, PG212), and the separated proteins were transferred to nitrocellulose membranes (Biosharp, Beijing, China, ISE00010). The membranes were blocked with blocking buffer (Shanghai Yase Biomedical Technology Co., LTD, Shanghai, China, PS108P) for 1 h. In addition, the membranes were incubated with antibodies against BAX, Bcl-2, GAPDH, and Tublin (β-tublin) (shown in Table 2) overnight at 4 °C. The membranes were coincubated with suitable secondary antibodies for an additional 1 h at room temperature. The bands were visualized with enhanced chemiluminescence (ECL) reagent (Biosharp, Beijing, China, BL520B).

### 2.13. Statistical Analyses

All experimental data in this article are expressed as mean ± standard deviation (SD). Origin 24.0 (OriginLab, Northampton, MA, USA) was used for statistical analysis and graph drawing. One-way analysis of variance (ANOVA) was performed to determine the differences among groups, and a *p*-value < 0.05 was considered statistically significant. The fluorescence intensity was analyzed with ImageJ software (1.8.0, National Institutes of Health, Bethesda, MD, USA), and the images were processed with Adobe Photoshop 2020 (San Jose, CA, USA).

## 3. Result

### 3.1. Establishment of Oxidative Stress Model

The treatment conditions in this experiment were investigated to establish the H_2_O_2_-induced antler MSCs oxidative stress model. Firstly, it can be seen from Figure 1A that the pluripotency marker octamer-binding transcription factor 3/4 (Oct3/4) and MSC surface markers, including CD90 and CD73, were significantly overexpressed in experimental cells compared with endometrial epithelial cells (EECs, dairy goats). The cells used in the experiment can be considered MSCs. The results showed that H_2_O_2_ treatment at ≥300 μmol/L significantly reduced the cell viability in a dose-dependent manner. Among them, the 2 h H_2_O_2_ (400 μmol/L) treatment significantly reduced the cell viability by approximately 30% (Figure 1B). qPCR was performed to detect the effect of 400 μmol/L H_2_O_2_ treatment for 2 h on the expression of MSC genes. The results showed that H_2_O_2_ treatment significantly up-regulated apoptosis-related genes *BAX* (Bcl2-associated X), *BAK* (Bcl-2 homologous antagonist/killer), and *caspase 7*. The expression levels of *SOD2* (superoxide dismutase 2), *CAT* (catalase), *GST* (glutathione-S-transferase), and *HO-1* were significantly down-regulated (Figure 1C,D). Meanwhile, ROS fluorescence intensity analysis showed that the 2 h H_2_O_2_ (400 μmol/L) treatment significantly increased ROS levels in MSCs (Figure 1E,F). In conclusion, a 2 h H_2_O_2_ (400 μmol/L) treatment can cause oxidative stress in antler MSCs. 

### 3.2. MLT Alleviating Oxidative Stress and Apoptosis of Antler MSCs

This study further evaluated the antioxidant and anti-apoptotic effects of MLT in antler MSCs. CCK-8 analysis showed that MLT significantly alleviated cell viability decline induced by H_2_O_2_ treatment (400 μmol/L), while 100 ng/mL MLT treatment had an optimal protective effect. MLT did not significantly alleviate cell viability decline induced by the treatment of H_2_O_2_ > 600 μmol/L (Figure 2A–D). The flow cytometry analysis showed that H_2_O_2_ (400 μmol/L) treatment significantly increased the S phase (DNA duplication phase) cell cycle proportion. MLT (100 ng/mL) treatment also significantly increased the S phase cell cycle proportion, but significantly decreased the G1 phase cell cycle proportion (Figure 2E,F). In addition, MLT treatment significantly inhibited H_2_O_2_-induced increase in intracellular ROS and MDA levels (Figure 2G,H). MLT treatment significantly inhibited the decrease of intracellular ATP levels induced by 200 μmol/L H_2_O_2_ treatment (Figure 2N). Furthermore, MLT treatment significantly inhibited the decrease of mitochondrial copy number and MMP induced by the 400 μmol/L H_2_O_2_ treatment (Figure 2J,K). MLT treatment significantly inhibited H_2_O_2_-induced early MSC apoptosis and expression of apoptotic gene BAX (Figure 2I,M,N). Thus, MLT can protect mitochondrial function and inhibit cell apoptosis by inhibiting H_2_O_2_-induced oxidative stress and cell damage in antler MSCs.

### 3.3. MLT Inhibiting H_2_O_2_-Induced Oxidative Stress in Antler MSCs through Receptor Pathway

In this article, luzindole—commonly used as a melatonin receptor inhibitor—was used to block the function of the receptor. The inhibitory effect of MLT on the receptor blockade of oxidative stress was studied. As shown in Figure 3A, the cell viability was significantly decreased after adding >1 μmol/L luzindole for 2 h compared with that before the receptor blockade; 10 μmol/L luzindole had the optimal inhibitory effect. The subsequent experiments showed that the receptor blockade significantly increased intracellular ROS and MDA levels (Figure 3B–D) and significantly reduced the MMP and mitochondrial copy number compared with the MLT + H_2_O_2_ group (Figure 3E–G). It was indicated that MLT inhibits H_2_O_2_-induced oxidative stress in antler MSCs and protects mitochondrial function through the melatonin receptor pathway.

### 3.4. MLT Activating Nrf2 Signaling Pathway and Inhibiting H_2_O_2_-Induced Apoptosis through MLT Receptor Pathway

qPCR was conducted to quantify the expression of Nrf2 pathway-related and apoptosis-related genes after treatment with H_2_O_2_, MLT, and luzindole. The results showed that H_2_O_2_ treatment significantly increased the expression of Nrf2-related genes (Figure 4A). MLT treatment further significantly up-regulated expression levels of genes, including *Nrf2*, *Keep1*, *NQO1* and *HO-1* compared with H_2_O_2_ treatment. Luzindole treatment significantly down-regulated the expression of these genes compared with MLT + H_2_O_2_ treatment (Figure 4B). Western blot was conducted to quantify the expression apoptosis-related genes after treatment with H_2_O_2_, MLT, and luzindole. The results showed that H_2_O_2_ treatment significantly increased the expression of *BAX* and decreased the expression of B-cell lymphoma-2 (*Bcl-2*). MLT treatment significantly down-regulated expression levels of *BAX* and up-regulated expression levels of *Bcl-2* compared with H_2_O_2_ treatment. As we expected, the addition of luzindole reversed this trend compared with the MLT + H_2_O_2_ group (Figure 4C–F).

## 4. Discussion

During the establishment of the antler MSCs oxidative stress model, H_2_O_2_ treatment reduced cell viability, down-regulated gene expression levels with antioxidant enzymes, and increased contents of intracellular ROS and MDA, indicating that the MSCs entered an oxidative stress state. Previous studies showed that MLT can alleviate oxidative stress [27], and our results showed that 100 ng/mL MLT had the most significant protective effect on H_2_O_2_-induced oxidative stress of antler MSCs. However, with increased H_2_O_2_ concentrations, the protective effect of MLT continued to decline until it disappeared. In addition, MLT exhibited different protective effects on various phenotypes of oxidative stress induced by gradient H_2_O_2_ treatments. MLT inhibited the increase of intracellular ROS induced by H_2_O_2_ at 500 μmol/L and below but had no significant inhibitory effect on the decrease of intracellular ATP induced by H_2_O_2_ at 300 μmol/L and above. Our results align with previous reports that, in cells under oxidative stress, ROS levels increase, mitochondrial function is damaged, and ATP levels decrease since mitochondria are the main site of ATP production [28]. 

The mitochondrial apoptosis pathway is regulated by the Bcl2 protein family and is the main pathway of cell self-elimination [29]. This Bcl2 protein family controls the permeability of the mitochondrial membrane and promotes the release of Cyt C into the cytoplasm. In the cytoplasm, Cyt C binds first with ATP and then with apoptotic protease activator (Apaf1) to form apoptotic body complex, thus, activating downstream elements of the caspase cascade [30]. Oxidative stress can trigger cell apoptosis [31]. In our study, H_2_O_2_ significantly increased intracellular oxidative stress, reduced the expression of genes with antioxidant enzymes, and up-regulated the expression of apoptotic genes, including *BAX, BAK*, and *CASP7*—which is consistent with the results of previous studies [32,33]. MLT is reported to be synthesized in the mitochondria of oocytes [34] and tends to accumulate in mitochondria [35]. MLT can alleviate oxidative stress by protecting mitochondrial function [36]. The results of our study further confirmed that MLT could significantly inhibit the H_2_O_2_-induced decline of mitochondrial copy number and MMP and reduce the proportion of cells that enter early apoptosis. Our results suggest that MLT in antler MSCs inhibits cell apoptosis by protecting mitochondrial function.

Exogenous melatonin can be absorbed by the body in a variety of ways, but the poor water-solubility of melatonin makes it unsuitable to be directly added into feed as a nutritional supplement [37]. Compared with the dietary supplement method, intramuscular injection can accurately control the amount of melatonin used, but it requires a higher frequency of operation and may cause greater stress to deer [38]. In the preliminary work of the laboratory, we used the method of subcutaneous implantation to study the effect of exogenous melatonin on the growth of antler velvet and achieved positive results, but the dosage and other treatment conditions of this method still need to be improved. The melatonin receptor plays an important role in MLT physiological function. MLT exerts antioxidant and anti-apoptotic effects through the melatonin receptors in follicular granulosa cells [39]. However, MLT is reducible and may undergo redox reactions with added H_2_O_2_. In order to reveal the protective mechanism of MLT in antler MSCs, the melatonin receptor inhibitor luzindole (10 μmol/L) was used to block the MLT receptor signal transduction (Luz + MLT + H_2_O_2_ group). The results showed that intracellular ROS and MDA were significantly increased, and the mitochondrial copy number and mitochondrial membrane potential were significantly decreased after the receptor signal blockade compared with those in the MLT + H_2_O_2_ group. The results indicated that MLT in antler MSCs can protect against H_2_O_2_-induced oxidative stress through the melatonin receptor.

The antioxidant enzyme system is important for the maintenance of intracellular redox balance. During the maturation of porcine oocytes, MLT can bind with melatonin receptor 2 to activate the Nrf2 signaling pathway, thus, resulting in the up-regulation of CAT [40]. Our data demonstrated that H_2_O_2_ treatment reduced the expression of antioxidant enzymes, and MLT treatment activated the Nrf2 pathway and inhibited apoptosis. MLT receptor blockage by luzindole significantly down-regulates the expression of Nrf2-related genes and up-regulated the expression of apoptosis-related genes. Thus, MLT activates the Nrf2 signaling pathway through the MTR, thereby, increasing the expression of genes with antioxidant enzymes and eventually playing an antioxidant and anti-apoptotic role. The results of previous studies showed that MLT could significantly reduce the levels of NO (nitric oxide) and ROS through the p38MAPK/NADPH oxidase signaling pathway [41]. Melatonin regulates the regeneration of periodontal ligament stem cells by activating PI3K/Akt/mTOR signaling pathway through melatonin receptor [42]. However, the mechanism of intracellular oxidative stress and apoptosis is very complex. Therefore, the antioxidant mechanism of MLT in antler MSCs needs to be further studied.

In conclusion, this study investigated the antioxidant and anti-apoptotic effects of MLT in antler MSCs. The results confirmed that MLT activates the Nrf2 pathway, elevates the expression of genes with antioxidant enzymes through the melatonin receptor, and alleviates the intracellular oxidative stress, thus, protecting the copy number and function of mitochondria and reducing the apoptosis and damage of antler MSCs. This study provides a theoretical basis for the oxidative stress and antioxidant process of antler MSCs and a reference for antler yield increase.

## Figures and Tables

**Figure 1 molecules-27-02515-f001:**
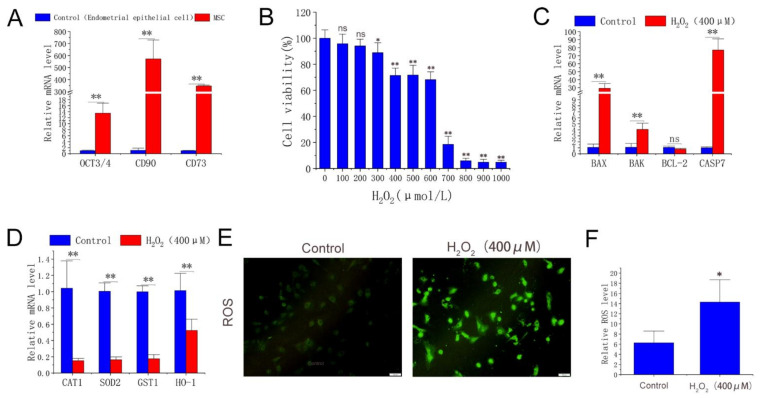
Cell type identification, effects of H_2_O_2_ on cell viability, gene expression, and ROS level. (**A**) mRNA level of *OCT3/4*, *CD90*, and *CD73* in experimental cells and EECs. (**B**) Cell viability determined by cck-8 method after gradient H_2_O_2_ treatment with 6 replicates per treatment. (**C**,**D**) Expression of apoptosis-related genes and antioxidant enzymes detected by qPCR with 3 replicates per treatment. (**E**) ROS fluorescence images captured by fluorescence-forward microscopy (n = 3, scale: 100 μm). (**F**) Fluorescence intensity calculated by ImageJ. H_2_O_2_ group was treated with H_2_O_2_ (400 μmol/L) for 2 h. The results were obtained from three independent experiments, and data were expressed as mean ± SD. * *p* < 0.05 (** *p* < 0.01) vs. control group; ns, not significant, *p* > 0.05 vs. control group.

**Figure 2 molecules-27-02515-f002:**
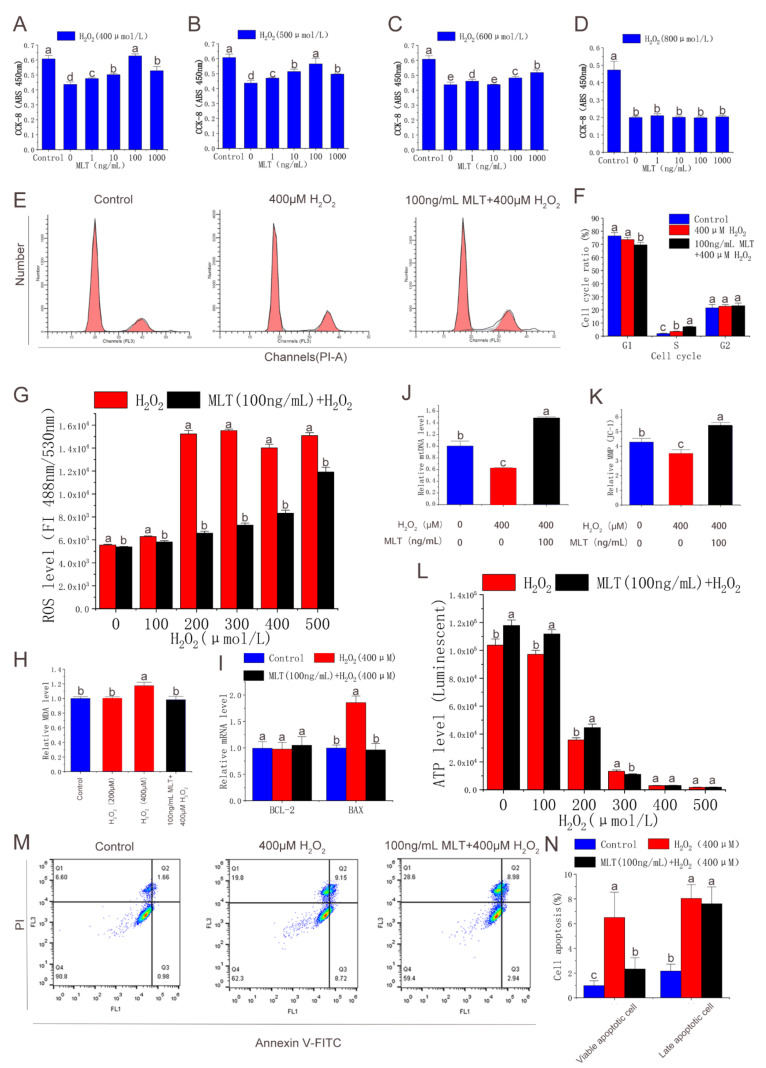
Effects of MLT treatment on cell viability, cell cycle, apoptosis, intracellular ROS, MDA level, and mitochondrial function. (**A**) Protective effect of gradient MLT on cell viability detected by CCK-8 method after treatment with concentrations of 400 μmol/L H_2_O_2_, n = 6. (**B**) Protective effect of gradient MLT on cell viability detected by CCK-8 method after treatment with concentrations of 500 μmol/L H_2_O_2_, n = 6. (**C**) Protective effect of gradient MLT on cell viability detected by CCK-8 method after treatment with concentrations of 600 μmol/L H_2_O_2_, n = 6. (**D**) Protective effect of gradient MLT on cell viability detected by CCK-8 method after treatment with concentrations of 800 μmol/L H_2_O_2_, n = 6. (**E**,**F**) Flow cytometry analysis of cell cycle in control group, H_2_O_2_ treatment group, and MLT + H_2_O_2_ group, n = 3. (**G**) Intracellular ROS level in gradient H_2_O_2_ group and 100 ng/mL MLT and gradient H_2_O_2_ group, n = 6. (**H**) MDA level under gradient H_2_O_2_ treatment and MLT + H_2_O_2_ treatment, n = 6. (**I**,**J**) Apoptosis-related gene expression and mtDNA levels in control group, H_2_O_2_ treatment group, and MLT + H_2_O_2_ group, n = 3. (**K**) Mitochondrial membrane potential in control group, H_2_O_2_ treatment group, and MLT + H_2_O_2_ group, n = 6. (**L**) Intracellular ATP level in control group, H_2_O_2_ treatment group, and MLT + H_2_O_2_ group, n = 6. (**M**,**N**) Flow cytometry analysis of cell apoptosis in control group, H_2_O_2_ treatment group, and MLT + H_2_O_2_ group, n = 3. H_2_O_2_ group, 2 h 400 μmol/L H_2_O_2_ treatment; MLT+ H_2_O_2_ group, 12 h 100 ng/mL MLT treatment, followed by MLT wash with PBS and 2 h 400 μmol/L H_2_O_2_ treatment. The results were obtained from three independent replicates. The same letters on the error bar indicate no significant difference between the groups, and different letters above error bars (a–e) represent significant differences between groups at the level of *p* < 0.05.

**Figure 3 molecules-27-02515-f003:**
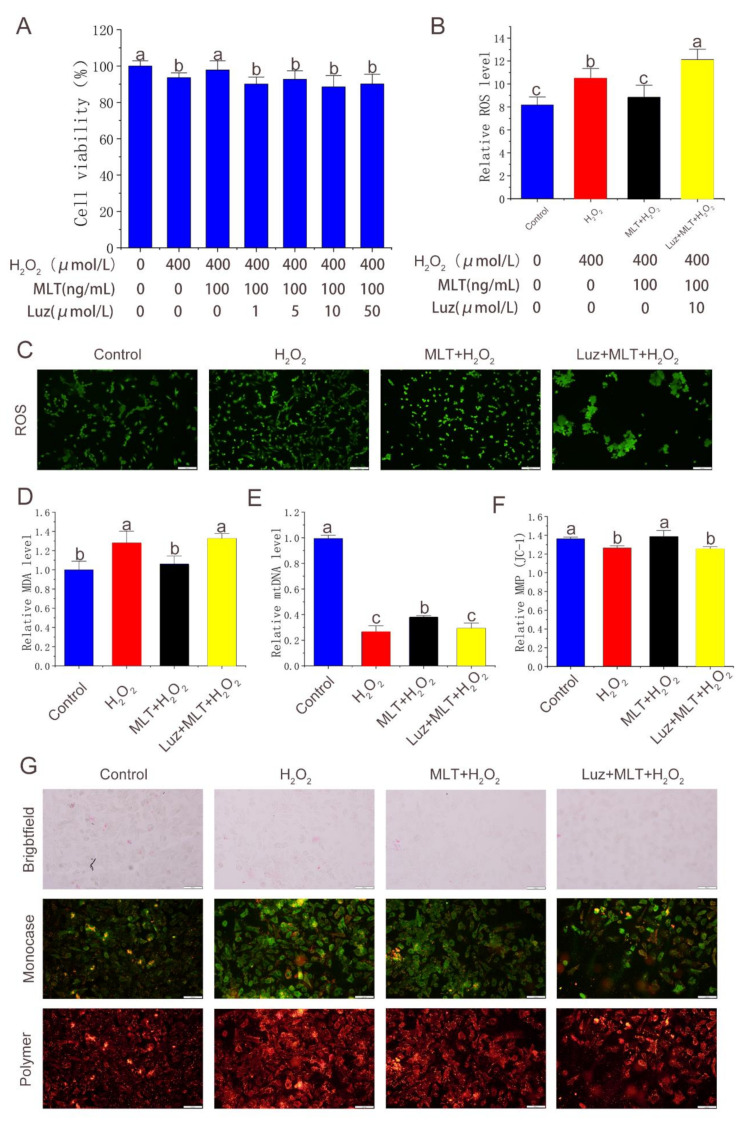
Effect of melatonin receptor blockade on alleviation of oxidative stress in antler MSCs by MLT. (**A**) Cell viability detected by CCK-8 method in control group, H_2_O_2_ treatment group, MLT+H_2_O_2_ group and Luz+ MLT+H_2_O_2_ group, n = 6. (**B**) Fluorescence intensity calculated by image-J in control group, H_2_O_2_ treatment group, MLT+H_2_O_2_ group and Luz+ MLT+H_2_O_2_ group, n = 3. (**C**) ROS fluorescence images captured by fluorescence forward microscopy in control group, H_2_O_2_ group, MLT+H_2_O_2_ group and Luz+ MLT+H_2_O_2_ group (scale: 200 μm). (**D**) Relative MDA content in control group, H_2_O_2_ group, MLT+H_2_O_2_ group and Luz+ MLT+H_2_O_2_ group. (**E**) Relative mtDNA expression level in control group, H_2_O_2_ group, MLT+H_2_O_2_ group and Luz+ MLT+H_2_O_2_ group, n = 3. (**F**) Relative MMP calculated by image-J in control group, H_2_O_2_ group, MLT+H_2_O_2_ group and Luz+ MLT+H_2_O_2_ group, n = 3. (**G**) MMP Fluorescence images captured by fluorescence forward microscopy in control group, H_2_O_2_ group, MLT+H_2_O_2_ group and Luz+ MLT+H_2_O_2_ group (scale: 200 μm). H_2_O_2_ group, 2 h 400 μmol/L H_2_O_2_ treatment; MLT + H_2_O_2_ group, 12 h 100 ng/mL MLT treatment, followed by MLT wash with PBS and 2 h 400 μmol/L H_2_O_2_ treatment, The results were obtained from three independent replicates. The data were expressed as mean ± SD. The same letters on the error bar mean no significant difference between groups, and different letters above error bars (a–c) represent significant differences between groups at the level of *p* < 0.05.

**Figure 4 molecules-27-02515-f004:**
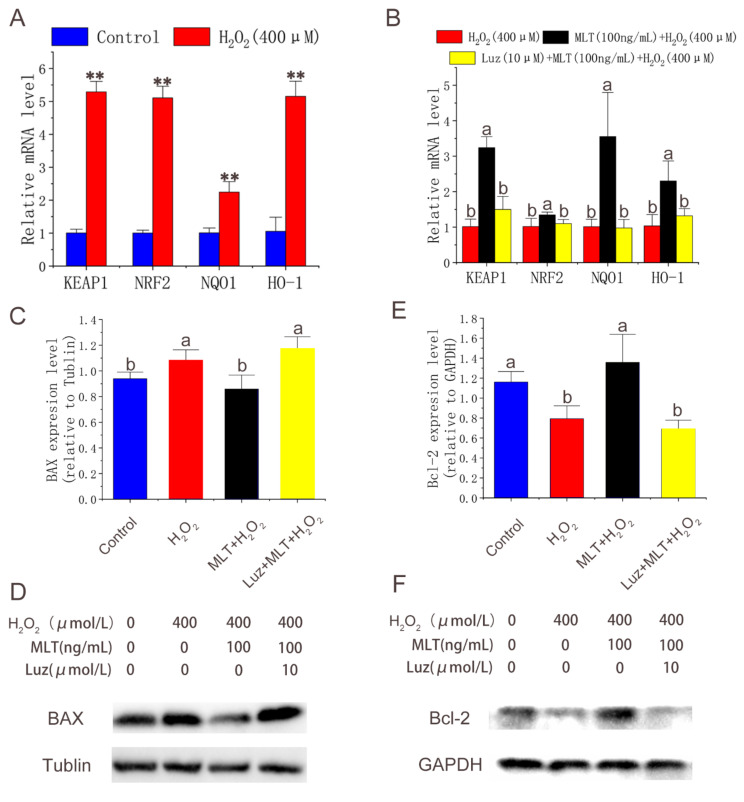
Expression of Nrf2 pathway-related and apoptosis-related genes. (**A**,**B**) Nrf2 pathway related genes expression level detected by qPCR in control group, H_2_O_2_ group, MLT+H_2_O_2_ group and Luz+ MLT+H_2_O_2_ group (n = 3). (**C**–**F**) BAX, Bcl-2 expression level detected by Western blot in control group, H_2_O_2_ group, MLT+H_2_O_2_ group and Luz+ MLT+H_2_O_2_ group. H_2_O_2_ group, 2 h 400 μmol/L H_2_O_2_ treatment; MLT + H_2_O_2_ group, 12 h 100 ng/ mL MLT treatment, followed by MLT wash with PBS and 2 h 400 μmol/L H_2_O_2_ treatment; Luz + MLT + H_2_O_2_ group, 2 h 10 μmol/L luzindole treatment, followed by 12 h 100 ng/ mL MLT treatment, Luz and MLT wash with PBS, and 2 h 400 μmol/L H_2_O_2_ treatment. The results were obtained from three independent replicates. The data were expressed as mean ± SD. ** *p* < 0.01 vs. the control group. The same letters on the error bar mean no significant difference between the groups, and different letters above error bars (a, b) represent significant differences between groups at the level of *p* < 0.05.

**Table 1 molecules-27-02515-t001:** Primer sequences for qPCR.

Genebank No.	Gene	Primer Sequence (5′-3′)	Amplicon Length (bp)
NM-001034034.2	GAPDH-F	GGAGTCCACTGGCGTCTTCA	240
	GAPDH-R	GTCATGAGTCCTTCCACGATACC	
NM-001101142	KEAP1-F	TACCTGGAGGCCTACAACCC	144
	KEAP1-R	GGTGTTACCATCAGGCGAGT	
NM-001011678.2	Nrf2-F	GCATGATGGACTTGGAGCTG	144
	Nrf2-R	GCTCATGCTCCTTCTGTCGT	
NM-001014912	HO-1F	CAAGCGCTATGTTCAGCGAC	200
	HO-1R	GCTTGAACTTGGTGGCACTG	
NM-001034535.1	NQO1-F	GGTGCTCATAGGGGAGTTCG	281
	NQO1-R	CCAGGCGTTTCTTCCATCCT	
NM-177516	GST1-F	AAGTTCCAGGACGGAGACCT	181
	GST1-R	CCGCCTCGTAGTTGGTGTAA	
XM-004016396	CAT1-F	CAGCCAGCGACCAGATGAAAC	277
	CAT1-R	ACCTTCGCCTTGGAGTATCTG	
NM-174615	SOD1-F	CATGTTGGAGACCTGGGCAA	148
	SOD1-R	CTCTGCCCAAGTCATCTGGTT	
NM-001280703	SOD2-F	TGTTGGTGTCCAAGGTTCCG	145
	SOD2-R	ATGCTCCCACACGTCAATCC	
NM-001126352.1	BCL-2F	GCTCTGGTGCTGGGTTATGA	291
	BCL-2R	CACTTTAGCCGAGGAGCAGG	
XM-027978593.1	BAX-F	CAGAGGCGGGTTTCATCC	279
	BAX-R	GCTGCAAAGTAGAAAAGGGC	
NM-001077918	BAK-F	GTCTTCCGCAGCTACGTCTT	104
	BAK-R	TGCTAGGTTCTGGGTGCAAG	
XM-002698509	CASP7-F	GTTGATGCAAAGCCAGACCG	263
	CASP7-R	CTCACATCGAAACCCAGGCT	
NM-174580	OCT-3/4-F	GTTTTGAGGCTTTGCAGCTC	182
	OCT-3/4-R	CTCCAGGTTGCCTCTCACTC	
NM-001034765	CD90-F	GTGAACCAGAGCCTTCGTCT	201
	CD90-R	GGTGGTGAAGTTGGACAGGT	
NM-174129.3	CD73-F	GTGTCGTGTGCCCAGTTATG	90
	CD73-R	AATCCGTCTCCACCACTGAC	
NC-006853.1	mtDNA-F	CTAAGCAGCCCGAAACCAGA	131
	mtDNA-R	ACAACCAGCTATCACCAGGC	

**Table 2 molecules-27-02515-t002:** The information of antibodies used for Western blot.

Antibodies	Cat NO.	Source	Dilution
BAX	AF0120	Affinit	1:1000
Bcl-2	Af6139	Affinit	1:1000
GAPDH	K106389P	Solarbio	1:1000
β-tublin	K106392P	Solarbio	1:1000
Goat anti-rabbit IgG	BL003A	Biosharp	1:5000

## Data Availability

Data will be available upon request to the corresponding author.

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
