# Peer review of "Exogenous Melatonin Activating Nuclear Factor E2-Related Factor 2 (Nrf2) Pathway via Melatonin Receptor to Reduce Oxidative Stress and Apoptosis in Antler Mesenchymal Stem Cells"

_molecules, 2022, doi:10.3390/molecules27082515_

Round 1

Reviewer 1 Report

Jing and coworkers investigated the antioxidant effect of melatonin over mesenchymal stem cells derived from deer antler. Even though the antioxidant property of melatonin has been widely reported on literature, its application on different areas is always a relevant research topic. Overall, the manuscript is well written, though a careful revision is recommended before its publication for avoiding long sentences and excess of article employments (especially “the”). Therefore, the manuscript could be considered for publication on Molecules if authors address the following major & minor concerns:

Major concerns

1- Lines 31-32: authors mentioned that antlers are of great economic and biological value. Authors should feed readers with more explanation/examples regarding the biological potential of antlers (supported by references).

2- Section 2.1., Lines 68-71: What is the specie/race related to deer that provided the antler? Authors should set up these information since they are relevant for the research made.

3- I strongly recommend that authors beef up the discussion on how melatonin would be beneficial for increasing antler yields. How melatonin would be administered? Oral formulation? As a nutritional supplement? Is there any research that supports melatonin use for similar purposes? What is still missing to advance in the field?

Minor concerns

1- Abstract, Lines 15, 19, and 25: Please, replace “H2O2” by “H2O2.

2- Abstract, Lines 18-19: the following sentence “…receptor blockade resulted in the significant increase of H2O2-induced hyperoxide levels and the significant inhibition of mitochondrial function.” should be “…receptor blockade resulted in significant increase of H2O2-induced hyperoxide levels and significant inhibition of mitochondrial function.”

3- Line 34: please replace “ultraviolet” by “ultraviolet light“.

4- Line 53: replace “…are combined in the cytoplasm and remain an inactive state.” by “…are combined in the cytoplasm, thus remaining in the inactive state.”

5- Line 58: “…have attracted the increasing attention…” should be “…have attracted increasing attention…”

6- Line 149: Please double check the up-regulation of “Csapcase 7” and “Csapcase 9”. Did you mean caspase 7 and 9?

7- Figure 2 has a poor resolution. Authors should increase Figure 2 quality.

8- Figure 3 idem.

9- Please standardize all acronyms and abbreviation (e.g. “Cyt C” or “CytC”).

Author Response

Major concerns

1- Lines 31-32: authors mentioned that antlers are of great economic and biological value. Authors should feed readers with more explanation/examples regarding the biological potential of antlers (supported by references).

Response 1: Thank you for your advice. We add a discussion of the biological value of velvet antler in lines 31-33, based on three new articles (in red).

2- Section 2.1., Lines 68-71: What is the specie/race related to deer that provided the antler? Authors should set up these information since they are relevant for the research made.

Response 2: Thank you for your advice. We have added basic information about the deer that provided the antler sample in lines 71-72 (in red).

3- I strongly recommend that authors beef up the discussion on how melatonin would be beneficial for increasing antler yields. How melatonin would be administered? Oral formulation? As a nutritional supplement? Is there any research that supports melatonin use for similar purposes? What is still missing to advance in the field?

Response 3: Thank you for your advice. We added a discussion on the use of melatonin and the previous experimental results in line 305-312 of the article (in red).

Minor concerns

1- Abstract, Lines 15, 19, and 25: Please, replace “H2O2” by “H2O2”.

2- Abstract, Lines 18-19: the following sentence “…receptor blockade resulted in the significant increase of H2O2-induced hyperoxide levels and the significant inhibition of mitochondrial function.” should be “…receptor blockade resulted in significant increase of H2O2-induced hyperoxide levels and significant inhibition of mitochondrial function.”

3- Line 34: please replace “ultraviolet” by “ultraviolet light“.

4- Line 53: replace “…are combined in the cytoplasm and remain an inactive state.” by “…are combined in the cytoplasm, thus remaining in the inactive state.”

5- Line 58: “…have attracted the increasing attention…” should be “…have attracted increasing attention…”

6- Line 149: Please double check the up-regulation of “Csapcase 7” and “Csapcase 9”. Did you mean caspase 7 and 9?

7- Figure 2 has a poor resolution. Authors should increase Figure 2 quality.

8- Figure 3 idem.

9- Please standardize all acronyms and abbreviation (e.g. “Cyt C” or “CytC”).

Response 4: Thank you for all your suggestions. We have modified them one by one in the article (in red).

Reviewer 2 Report

Dear Editor,

Thank you very much for an opportunity to review the manuscript entitled: “Exogenous melatonin activating nuclear factor E2-related factor 2 2 (Nrf2) pathway via melatonin receptor to reduce oxidative 3 stress and apoptosis in antler mesenchymal stem cell”. This is potentially interesting study; however authors have to clear some major concerns before publication.

Major comments:

  1. Please provide the evidence that experiments were performed on MSC (please add a figure with analysis of expression MSC markers (mRNA and protein).
  2. Please provide all necessary information to understand figure without the reading of the text in the legends. For example, concentration of the drugs. Please use staples to show statistical significance between groups. There is huge problem with interpretation of presented data based on description: Different lower-case letters above bars (a, b) represented significant differences between groups 242 at the level of P<0.05.

Some panels have letter up to e?

  1. Please provide information concerning the antlers (from what organism antlers were taken).
  2. How did you design the primers, did you confirmed the specificity of products by sequencing?
  3. Key results should be confirmed on protein level.

Minor points:

Please explain all abbreviations when they first appear (PM2.5)

Author Response

Major comments:

  1. Please provide the evidence that experiments were performed on MSC (please add a figure with analysis of expression MSC markers (mRNA and protein).

Response 1: Thanks for your suggestion, we added the identification of the expression level of multipotent gene POU5F1 in deer antler mesenchymal cells and chondrocytes (Figure 1D). In addition, cell types were identified by toluidine blue, cyrene S and Alcian blue staining in previous experiments.

Unfortunately, this figure has already been published, so it cannot appear in this article. As shown in the figure below:

  1. Please provide all necessary information to understand figure without the reading of the text in the legends. For example, concentration of the drugs. Please use staples to show statistical significance between groups. There is huge problem with interpretation of presented data based on description: Different lower-case letters above bars (a, b) represented significant differences between groups 242 at the level of P<0.05. Some panels have letter up to e?

Response 2: Thank you for your advice. We modified the expression forms of partial significance in FIG. 1 and FIG. 4. However, in the bar chart with more groups, we retained the original expression forms in order to more intuitively express the significant differences between groups. As for the problem you mentioned that each treatment should have a clear dose in the diagram, we have also made optimization (in red).

  1. Please provide information concerning the antlers (from what organism antlers were taken).

Response 3: Thank you for your advice. We have added basic information about the deer that provided the antler sample in lines 71-72 (in red).

  1. How did you design the primers, did you confirmed the specificity of products by sequencing?

Response 4: Thank you for your advice. Our primer design was carried out through NCBI primer BLAST. After the primers were designed, PCR and agarose gel electrophoresis were performed. The criterion was that the bands of gel electrophoresis were single and bright and consistent with the expected size.

  1. Key results should be confirmed on protein level.

Response 5:Thank you for your advice. During this period, we supplemented the effects of each treatment on the protein level of apoptotic genes, and the experimental results were shown in lines 258-264 (Figure 4), which met our expectations.

Minor points:

Please explain all abbreviations when they first appear (PM2.5)

Response 6:Thank you for your advice. We have corrected similar problems.

Round 2

Reviewer 2 Report

Dear Authors,

Thank you very much for detail revision of the manuscript.

Unfortunately, few serious issues have to be clarity before publication.

Major comments:

  1. Please provide the evidence that experiments were performed on MSC (please add a figure with analysis of expression MSC markers (mRNA and protein).

One marker with 1.8 fold change studied by qPCR is not enough.

The paper concerning MSC (citation 26) concerns human cells, so may not apply to deer.

  1. Please provide information concerning the antlers (from what organism antlers were taken). Please note that Cervus nippon has several subspecies.

  1. How did you design the primers, did you confirmed the specificity of products by sequencing?

Once again, I have not found any full genome sequence of Cervus nippon,

So how those primers were design? Based on what organism DNA sequence?

How do you know what is the length of the PCR product in Cervus nippon?

You have to confirm specificity of PCR products by sequencing or provide convincing explanation.

Author Response

Major comments:

  1. Please provide the evidence that experiments were performed on MSC (please add a figure with analysis of expression MSC markers (mRNA and protein).

One marker with 1.8 fold change studied by qPCR is not enough.

Response 1: Thanks for your suggestion. In this paper to modify, we identified mesenchymal cells used in the experiment by referring to the identification method of mesenchymal cells in " Bovine tongue epithelium-derived cells: A new source of bovine mesenchymal stem cells ". The experimental results showed that compared with endometrial epithelial cell (EEC) in dairy goats, the expression level of OCT3/4 in mesenchymal cells was more than ten times that of EEC, and the expression level of CD90 and CD70 was hundreds of times that of EEC, which could confirm that the cells used in our experiment had mesenchymal cell characteristics.

The paper concerning MSC (citation 26) concerns human cells, so may not apply to deer.

  1. Please provide information concerning the antlers (from what organism antlers were taken). Please note that Cervus nipponhas several subspecies.

Response 2: Thanks for your suggestion. The Cervus nippon used in this experiment is shuangyang sika deer. The scientific research topic of "Shuangyang Sika deer Breeding" began in the 1960s, presided over by Han Kun, the former senior animal husbandry division of the third State-owned Shuangyang Deer Farm. In 1979, the research project was listed as one of the key research projects of the Ministry of Agricultural Reclamation of China. In 1986, the "Shuangyang Sika deer breeding" project achieved fruitful results. After more than 20 years of artificial cultivation, a new variety of sika deer, "Shuangyang Sika Deer", passed the technical appraisal organized by the State Ministry of Agricultural Reclamation.

  1. How did you design the primers, did you confirmed the specificity of products by sequencing?

Once again, I have not found any full genome sequence of Cervus nippon,

So how those primers were design? Based on what organism DNA sequence?

How do you know what is the length of the PCR product in Cervus nippon?

You have to confirm specificity of PCR products by sequencing or provide convincing explanation.

Response 3: Thanks for your suggestion. I am sorry that I did not understand your suggestion accurately before. Here I would like to explain that the design of primers is based on the same gene of cattle or sheep (cattle and sheep are closely related to deer). Your suggestion is very necessary. We found that the design of the two primers CASP9 and GPX were indeed problematic during base sequence sequencing. The experimental results related to these two genes have been deleted. Thank you again for your suggestions. There was no problem in the design of other primers. BLAST was conducted in NCBI, and the homology was above 90%. The following the results of base sequencing.
